# The Haptoglobin Response after Aneurysmal Subarachnoid Haemorrhage

**DOI:** 10.3390/ijms242316922

**Published:** 2023-11-29

**Authors:** Soham Bandyopadhyay, Patrick Garland, Ben Gaastra, Ardalan Zolnourian, Diederik Bulters, Ian Galea

**Affiliations:** 1Clinical Neurosciences, Clinical & Experimental Sciences, Faculty of Medicine, University of Southampton, Southampton SO16 6YD, UK; s.bandyopadhyay@soton.ac.uk (S.B.); p.garland@kedrion.com (P.G.); b.gaastra@soton.ac.uk (B.G.); 2Wessex Neurological Centre, University Hospital Southampton NHS Foundation Trust, Southampton SO16 6YD, UK; ardalan.zolnourian@uhs.nhs.uk

**Keywords:** subarachnoid haemorrhage, cerebrospinal fluid, haptoglobin, haemoglobin, cytokines

## Abstract

Haptoglobin is the body’s first line of defence against the toxicity of extracellular haemoglobin released following a subarachnoid haemorrhage (SAH). We investigated the haptoglobin response after SAH in cerebrospinal fluid (CSF) and serum. Paired CSF and serum samples from 19 controls and 92 SAH patients were assayed as follows: ultra-performance liquid chromatography for CSF haemoglobin and haptoglobin, immunoassay for serum haptoglobin and multiplexed CSF cytokines, and colorimetry for albumin. There was marked CSF haptoglobin deficiency: 99% of extracellular haemoglobin was unbound. The quotients for both CSF/serum albumin (qAlb) and haptoglobin (qHp) were used to compute the CSF haptoglobin index (qHp/qAlb). CSF from SAH patients had a significantly lower haptoglobin index compared to controls, especially in Haptoglobin-1 allele carriers. Serum haptoglobin levels increased after SAH and were correlated with CSF cytokine levels. Haptoglobin variables were not associated with long-term clinical outcomes post-SAH. We conclude that: (1) intrathecal haptoglobin consumption occurs after SAH, more so in haptoglobin-1 allele carriers; (2) serum haptoglobin is upregulated after SAH, in keeping with the liver acute phase response to central inflammation; (3) haptoglobin in the CSF is so low that any variation is too small for this to affect long-term outcomes, emphasising the potential for therapeutic haptoglobin supplementation.

## 1. Introduction

Central to the toxicity of extravascular blood is the presence of haemoglobin [1,2]. Following subarachnoid haemorrhage (SAH), haemoglobin is released into the cerebrospinal fluid (CSF) due to the breakdown of red blood cells. Haemoglobin is thought to play a role in the secondary brain injury (SBI) that follows SAH through mechanisms including inflammation, free radical creation, nitric oxide consumption, and oedema [3,4,5]. Additionally, haemoglobin has been associated with tertiary processes, such as vasoconstriction, microthrombi formation, and iron deposition, which can themselves further exacerbate SBI [6,7]. Notably, microthrombi can obstruct micro-vessels [8] and disrupt the glymphatic drainage system [9].

Haptoglobin (Hp) is an abundant acute-phase glycoprotein—mainly produced by the liver—that binds to extracellular haemoglobin with high affinity [10]. Intrathecal Hp is blood-derived under normal conditions [11]. It can provide protection against haemoglobin-induced neurotoxicity [5,12,13,14,15,16] through various mechanisms: lowering the redox potential of haemoglobin to prevent damaging oxidative reactions; sequestering haemoglobin to limit its diffusion into brain tissue; and chaperoning haemoglobin out of the CSF, thereby preventing haem release and its degradation to free iron [17]. The Hp-haemoglobin complex is recognised and endocytosed by cells expressing the CD163 cell surface receptor [18]. The CD163 Hp-haemoglobin scavenging system—present at a low capacity in the central nervous system [19]—extends beyond clearing haemoglobin; it is also implicated in generating an anti-inflammatory response [20]. Therefore, there is a strong biological rationale to hypothesise that Hp may have therapeutic potential in reducing SBI in SAH.

While Hp is an abundant protein in plasma, it is present at much lower levels in the CSF of both healthy individuals and SAH patients [11,19]. CSF Hp levels are complex to interpret after SAH since variable amounts of Hp are released into the CSF at the time of the bleed, and consideration needs to be given to changes in serum Hp and blood–brain barrier (BBB) permeability before conclusions are made regarding intrathecal Hp synthesis and/or consumption. It is also not clear whether serum Hp production increases post-SAH, and whether there is neuro-immune communication from the brain to the periphery that drives this response.

An individual’s Hp phenotype can influence the level of CSF Hp [11,19]. In humans, there are two *HP* alleles: *HP1* and *HP2* [21,22]. The basic unit of Hp protein is a monomer consisting of an α and a β subunit. The α subunit in individuals with the *HP*1 allele has one cysteine residue, enabling Hp monomers to dimerise. In contrast, the α subunit in individuals with the *HP*2 allele has two cysteine residues, leading to polymers being formed from three or more Hp monomers [23,24,25]. An individual can have one of three genotypes: HP1-1 homozygote, HP2-1 heterozygote, and HP2-2 homozygote. The different Hp types bind haemoglobin with identical affinities but may have functional differences related to haemoglobin-binding capacity, protection against haemoglobin-induced neurotoxicity, binding of complexes to CD163, subsequent endocytosis, and the ability to generate anti-inflammatory responses [23,26]. Consequently, Hp turnover and function across different Hp phenotypes is a topic of significant interest.

This study aimed to understand the CSF and serum Hp response following SAH; hence, the primary endpoint was the Hp level in CSF and serum at day 7 in SAH patients compared to controls. Secondary endpoints included: (1) the differences between Hp phenotypes and (2) the association between Hp variables and long-term clinical outcomes after SAH.

## 2. Results

### 2.1. Demographics and Clinical Characteristics

A total of 92 patients with SAH recruited between April 2016 and February 2019 were included in our analysis. Most participants had the Hp 2-1 phenotype (n = 46/92, 50.0%). There were no significant associations between the baseline characteristics of SAH patients and their Hp phenotype (Table 1). In addition, 19 control participants were recruited: 3 (15.8%) were Hp 1-1, 9 were Hp 2-1 (47.4%), and 7 were Hp 2-2 (36.8%).

### 2.2. Evidence for Hp Consumption in the CSF

The median level of Hp in the CSF of patients 7 days after SAH was 0.002 g/L (IQR: 0.001–0.005 g/L). The median level of Hp in the CSF of 19 controls was similar at 0.003 g/L (IQR: 0.0001–0.006 g/L) (*p* = 0.850). However, the effect of Hp phenotype on CSF Hp levels differed between SAH and controls. While there was no difference in day 7 CSF Hp levels between patients with different Hp phenotypes (*p* = 0.714), CSF Hp levels in controls varied across different Hp phenotypes (*p* = 0.008) (Figure 1).

Among those with the Hp 1-1 phenotype, SAH patients [Median: 0.003 g/L; IQR: 0.001–0.007 g/L] had a significantly lower level of CSF Hp compared to controls (*p* = 0.044). Among those with the Hp 2-2 phenotype, SAH patients [Median: 0.002 g/L; IQR: 0.0007–0.004 g/L] had a significantly higher level of CSF Hp compared to controls (*p* = 0.047). There was no significant difference in CSF Hp levels between SAH patients [Median: 0.002 g/L; IQR: 0.001–0.005 g/L] and controls with the Hp 2-1 phenotype (*p* = 0.641).

These differences in CSF Hp levels between SAH patients and controls among participants with different Hp phenotypes reflect both an altered CSF Hp turnover after SAH and a significant effect of Hp phenotype on CSF Hp turnover. Since serum Hp levels and BBB permeability influence CSF Hp levels [11,27], the intrathecal Hp index was employed to account for serum Hp level and BBB permeability:(1)$$\frac{CSF\;haptoglobin÷serum\;haptoglobin}{CSF\;albumin÷serum\;albumin}.$$

After SAH, several factors may affect CSF Hp concentration (Figure 2). Since the intrathecal Hp index corrects for serum Hp levels and blood–brain barrier permeability, it becomes a measure of the balance between the two residual factors: intrathecal synthesis and CD163-mediated consumption of Hp. A higher Hp index after SAH, compared to controls, indicates that Hp intrathecal synthesis overtakes Hp consumption, while a lower Hp index indicates that Hp consumption predominates over Hp synthesis. The Hp index for all SAH patients was not significantly associated with blood volume (*p* = 0.317) or WFNS grade (*p* = 0.187). Considering just those patients with no intracerebral haemorrhage, the Hp index was still not significantly associated with blood volume (*p* = 0.982). This confirmed that the Hp index value on day 7 was not dependent on the initial blood volume entering the subarachnoid space.

The Hp index was significantly lower in SAH patients [Median: 0.083; IQR: 0.051–0.162] compared to controls [Median: 0.314; IQR: 0.053–0.495] (*p* = 0.016), and this was influenced by the Hp phenotype, such that the Hp index decreased in proportion to *HP*1 allele number (Figure 3). SAH patients with the Hp 1-1 phenotype [Median Hp index: 0.099; IQR: 0.075–0.246] had the largest decrease in Hp index compared to controls [Median: 1.220; IQR: 1.045–1.395] (*p* = 0.025). SAH patients with the Hp 2-1 phenotype [Median Hp index: 0.085; IQR: 0.048–0.162] also had a significantly lower Hp index compared to controls [Median: 0.384; IQR: 0.251–0.478] (*p* = 0.012), but the decrease was not as marked as in Hp 1-1 patients. There was no significant difference in Hp index between SAH patients (Median: 0.062; IQR: 0.047–0.127) and controls (Median: 0.053; IQR: 0.027–0.207) with the Hp 2-2 phenotype (*p* = 0.760).

### 2.3. Serum Hp Increases Following SAH

The median level of Hp in the serum of patients 7 days after SAH was 3.07 g/L (IQR: 2.41–3.60 g/L). The median level of Hp in the serum of 15 controls was 1.39 g/L (IQR: 1.14–1.94 g/L). This was significantly lower compared to SAH patients (*p* < 0.001).

SAH patients with the Hp 2-2 phenotype had lower day 7 serum Hp level [Median: 2.72 g/L; IQR: 1.98–3.23 g/L] compared to patients with Hp 2-1 phenotype [Median: 3.07 g/L; IQR: 2.50–3.77 g/L] (*p* = 0.023) and patients with the Hp 1-1 phenotype [Median: 3.19 g/L; IQR: 2.26–3.61 g/L] (*p* = 0.090) (Figure 4). This significant difference in day 7 serum Hp levels between Hp phenotypes remained when patients with the Hp 1-1 phenotype were combined with those of the Hp 2-1 phenotype [Median: 3.16 g/L; IQR: 2.42–3.71 g/L] and compared to the Hp 2-2 phenotype group (*p* = 0.045).

Controls with the Hp 2-2 phenotype had lower serum Hp levels [Median: 1.14 g/L; IQR: 0.94–1.55 g/L] compared to participants with the Hp2-1 phenotype [Median: 1.32 g/L; IQR: 1.23–2.00 g/L] (*p* = 0.098) and participants with Hp 1-1 phenotype [Median: 2.40 g/L; IQR: 1.91–2.89 g/L] (*p* = 0.019). Serum Hp levels were significantly greater in SAH patients compared to controls for those with Hp 2-1 phenotype (*p* < 0.001) or Hp 2-2 phenotype (*p* = 0.006), but not in those with Hp 1-1 phenotype (*p* = 0.119).

Since Hp is an acute phase reactant, the increase in serum Hp after SAH is in keeping with an acute phase response after SAH, as described in animal models of brain injury (Figure 5). Specifically, experimental injection of the cytokine IL-1beta within the brain in mice resulted in an acute phase response in the liver [28], which is the predominant source of circulating Hp production [29]. For this reason, we next investigated the relationship between CSF cytokines and serum Hp after SAH.

### 2.4. Serum Hp Levels Are Associated with CSF Cytokines

Serum Hp levels were significantly associated with CSF IL-1beta levels after SAH (β = 0.534 [95% CI: 0.071–0.997; *p* = 0.024]) when considering all Hp phenotypes. The association between CSF cytokines and serum Hp levels varied between Hp phenotypes (see Appendix A). For this reason, an analysis was conducted after adjusting for interaction with Hp phenotype (Table 2). This once again confirmed the positive association between serum Hp levels and CSF cytokine levels.

Since multiple cytokines act in concert in biological systems, a principal component analysis of CSF cytokines after SAH was conducted to identify the principal components with an eigenvalue greater than 1.0 (Figure 6). Table 3 shows the relationship between these three principal components and CSF cytokines. There was a significant association between serum Hp levels and the principal component with the highest eigenvector, when adjusting for interaction with Hp phenotype (Table 2).

### 2.5. Hp Is Not Associated with Outcome

We finally sought to determine whether key variables relating to Hp turnover had any long-term clinical correlates. In the univariate analysis, no significant association was found between serum Hp levels, Hp index, or Hp phenotype and the following outcome measures: death, DCI, NCI, Checklist for Cognitive and Emotional Consequences of Stroke (CLCE-24) [30], Extended Glasgow Outcome Scale (GOSE) [31], modified Rankin Scale (mRS) [32], and the SAH Outcome Tool (SAHOT) [33] at 90 or 180 days.

Since Hp was not associated with outcome, we next wanted to determine whether Hp was present in sufficient amounts to bind CSF haemoglobin levels after SAH. The median level of day 7 CSF haemoglobin was 4.65 µM (IQR: 0.58–18.42 µM) and did not differ by Hp phenotype (*p* = 0.872). Per patient, there was only sufficient Hp present in the CSF to bind a maximum of 1.08% (IQR: 0.38–3.82%) of the total haemoglobin in the CSF 7 days after SAH, and this did not differ by Hp phenotype (*p* = 0.325). This marked CSF Hp deficit may underlie the lack of association between Hp dynamics and clinical outcomes.

## 3. Discussion

### 3.1. Key Findings

A novel and striking finding is that SAH patients had a significantly lower CSF Hp index compared to controls, in keeping with intrathecal Hp consumption within the central nervous system (CNS) following SAH. Moreover, there was a significant effect of Hp phenotype on CSF Hp turnover, such that the decrease in Hp index varied in proportion to the *HP1* allele number. However, no significant association was identified between variables relating to Hp turnover and long-term clinical outcome following SAH. This is likely due to the levels of CSF Hp being insufficient to bind most of the extracellular haemoglobin in the CSF, with approximately 99% of extracellular haemoglobin remaining unbound. Hence, Hp is not present in sufficient amounts to effect outcome. A second important finding was that serum Hp levels increased after SAH and were significantly associated with CSF cytokine levels, providing the first human evidence of a systemic acute phase response to a central inflammatory stimulus.

### 3.2. Implications

Previous studies have described that, in healthy individuals, CSF Hp is higher in those with the Hp 1-1 phenotype compared to Hp 2-1 and Hp 2-2 [11,34]. This is consistent with our data. These previous studies have put down their findings to the differential permeability of the BBB to blood-derived Hp, as one clear difference between Hp phenotypes is the molecular weight of Hp. For example, in Hp 1-1 individuals, the Hp dimers have a molecular weight of 89 kDa, whilst Hp 2-2 individuals have Hp polymers that can range in molecular weight from 199 kDa to >1000 kDa. Bigger polymers have larger hydrodynamic radii, and there is an inverse relationship between hydrodynamic radii and BBB penetration [35]. In our control individuals, serum Hp levels were higher in proportion to the number of *HP*1 alleles. This finding is consistent with studies that have been conducted in a multitude of populations: European (Belgian [36,37], Iceland [38]), East Asian (Japanese [39], Koreans [40]), African (Zimbabweans [41], Gabonese [42]), and Melanesians [43].

The intrathecal Hp index is useful for studying intrathecal Hp dynamics in SAH patients since it takes into account changes in serum Hp and BBB permeability. A decrease in the Hp index compared to controls after SAH suggests that Hp consumption exceeds synthesis in the intrathecal compartment [44]. Hp consumption within the CNS after SAH would arise from uptake of Hp-haemoglobin complexes by CD163-positive macrophages [19] (Figure 2). The difference in Hp index between SAH patients and controls was observed to increase in proportion to the number of *HP*1 alleles. This could signify that CSF Hp consumption after SAH is highest for Hp 1-1, followed by Hp 2-1 and Hp 2-2 individuals. A functional difference in CD163 uptake between Hp types is supported by a study by Asleh et al. [45]. Using rhodamine-tagged and 125I-Hp in cell lines stably transfected with CD163 and in macrophages expressing endogenous CD163, they found that the rate of clearance of Hp 1-1-containing haemoglobin complexes by CD163 is markedly greater than that of Hp 2-2 [45].

Drainage of complexes from the brain may occur via endogenous pathways (such as the glymphatic system [46,47,48] and the intramural periarterial drainage pathway [49,50,51] or CSF absorption into the circulation via arachnoid granulations or into the nasal lymphatics via the cribriform plate [52]). These pathways are thought to be impaired after SAH [53]. More efficient clearance of Hp 1-1 haemoglobin complexes, versus those containing Hp 2-2, is intuitive since movement of interstitial solutes in the brain is dependent on molecular weight [48,49,54] and smaller Hp 1-1 complexes could therefore be able to more easily move along paravascular spaces. However, this cannot explain the decrease in Hp index in Hp 1-1 individuals compared to controls, which is therefore likely to reflect intrathecal CD163-mediated uptake.

Hp was detected in the CSF of SAH patients despite high levels of extracellular haemoglobin. This indicates saturation of the intrathecal CD163 scavenging system across Hp phenotypes as documented previously [19]. The likely reason our study did not find an association between Hp phenotype and clinical outcomes is because of a marked Hp deficit in the CNS, which is well recognised in the literature [19]. With intrathecal administration, CSF Hp levels will be higher, and functional differences between Hp phenotypes may become apparent [26]. Therefore, intrathecal haptoglobin administration has therapeutic potential.

We observed an upregulation of serum Hp level after SAH. Since Hp is an acute phase reactant produced by the liver and a brain–liver pathway underlying a systemic acute phase response was described after experimental intracerebral inflammatory challenge [28,55], we investigated the association between serum Hp and CSF cytokines following SAH. This provides the first human evidence of a systemic acute phase response to a central inflammatory stimulus (Figure 5). It was striking that of all the CSF cytokines, it was IL-1beta that was significantly positively associated with serum Hp levels in our results, as previous animal studies have shown an acute phase response in the liver after intracerebral injection of IL-1beta [28,55]. A time delay of six hours was reported between the injection and the systemic acute phase response [28], suggesting that there is a window of opportunity to target the signal relay between the brain and the liver. Given that the increase in brain cytokines is not associated with a corresponding increase of the same cytokines in the periphery, the local release of free cytokine from the CSF into the circulation to target peripheral organs is unlikely to be the major signalling pathway [55,56]. The signalling is thought to be mediated by extracellular vesicles (EVs). EVs are membrane-enclosed vesicles comprising larger microvesicles (100 nm–1 µm) formed by the outward budding and fission of the plasma membrane, and smaller exosomes (<200 nm) formed by the endocytic invagination of endosomal membranes and stored in multivesicular bodies [57]. EVs generated following intracerebral injection of IL-1beta in rats were successfully isolated and then transferred to naïve rats to activate a systemic acute phase response [58]. Similarly, EVs derived from brain endothelial cell cultures treated with IL-1beta also activated a systemic acute phase response in recipient animals [58]. EVs being a targeted inter-organ communication system provides a whole new avenue for the development of future therapies treating inflammation in SAH.

Also of interest is the induction of the rat homologue of IL-8 (Cytokine-induced neutrophil chemoattractant-1 (CINC-1)) in the liver following the intracerebral injection of IL-1beta, and neutralisation of CINC-1 in the periphery reverses neutrophil mobilisation and recruitment to the brain [55]. Given leucocyte mobilisation into the brain is associated with poorer outcomes after SAH [59], it would be of interest to determine the time-course of plasma IL-8 production in SAH patients and its relation to CSF leucocytosis. Animal studies have also shown that the intraparenchymal injection of TNF-alpha is associated with a systemic acute phase response and leucocytosis [56]. However, in our results, serum Hp levels were negatively associated with CSF TNF-alpha levels. This discrepancy might be because the relationship between TNF-alpha and acute phase protein production varies over time, and our study might have captured the relationship at a resolution phase during which tristetraprolin—a negative feedback inhibitor of TNF-alpha—is induced. This hypothesis merits further study by exploring the temporal trend of TNF-alpha, serum Hp, and tristetraprolin following SAH [60].

This study has several strengths and limitations. The sample size was substantial, with paired CSF and serum sampling on the same day after SAH. Day 7 is sufficiently distant from ictus to avoid any interference from the Hp entering the brain with the initial bleed—indeed the day 7 CSF Hp index was not associated with blood volume. On the other hand, limiting the analysis to one timepoint may not capture the dynamics of Hp. Roughly half the SAH patients on this study received SFX-01, which contains sulforaphane and has been shown in experimental studies to upregulate Hp transcription [61]. However, a sensitivity analysis just considering the patients who were on the placebo showed similar results (Hp index still lower, *p* = 0.002, and serum Hp still higher, *p* < 0.001, after SAH compared to controls). We used a sensitive ultra-performance liquid chromatography (UPLC) technique to measure Hp and extracellular haemoglobin in the CSF. Missing data for some patients could potentially introduce bias. While CSF samples were collected via an external ventricular drain (EVD) or lumbar puncture (LP), and this may be seen to introduce heterogeneity due to the rostro-caudal gradient in CSF protein content [62,63], this was minimised by the use of the Hp index, which is a ratio between Hp and albumin CSF/serum quotients. Finally, this study did not measure other factors in the CSF that have been associated with outcomes, such as glucose and lactate [64].

## 4. Materials & Methods

### 4.1. Study Design and Participants

The samples and data used in this study originate from a prospective, multicentre, randomised, double-blind, placebo-controlled study designed to investigate the safety and efficacy of SFX-01 (a synthetic sulforaphane stabilized within an α-cyclodextrin complex) in patients with SAH (NCT02614742). The SFX-01 after SAH (SAS) trial [65] was approved by the National Research Ethics Service South Central Hampshire A committee (ref no 16/SC/0019), and all patients or their legal representative gave consent to the storage and use of their samples and data for research. It is important to note that this paper does not address the efficacy of SFX-01 but rather investigates the Hp response after SAH, using samples and data from this trial. Clinical presentation and outcome data were present for 105 patients, who were originally recruited for the trial within 48 h of sustaining Fisher grade 3 or 4 SAH. Of these 105 patients, 92 were included in this study (Figure 7). CSF and plasma samples were obtained from 19 control individuals without SAH who participated in a study entitled “novel biomarkers in head injury” (National Health Service regional ethics committee approvals 11/SC/0204 and 10/H0502/53). These individuals were selected as controls as they had CSF with normal constituents obtained from a clinically indicated LP and were not subsequently diagnosed with an underlying inflammatory, neurodegenerative, or vascular neurological condition.

### 4.2. CSF and Plasma Sampling

Of the 92 SAH patients included for this study, 82 had CSF samples and 90 had serum samples taken. Of the 82 patients with CSF samples, 32 had CSF samples obtained at day 7 (±1 day) via an EVD sited for clinical hydrocephalus. The first 3 mL of CSF (representing dead space) was discarded to ensure fresh CSF was obtained. 50 patients without an EVD had an LP at day 7 (±1 day) and had CSF available for analysis. Day 7 follows the acute rise in Hp following SAH as a result of injection of blood into the subarachnoid space, after which it subsequently declines within the first few days to a relatively steady state by day 7 [16].

### 4.3. CSF and Plasma Cytokine Analysis

Cytokines—interferon-gamma (IFN-γ), IL-1beta, IL-10, IL-12p70, IL-13, IL-2, IL-4, IL-6, IL-8, and tumour necrosis factor alpha (TNF-α)—were quantified using the Meso Scale Discovery (MSD) Multi-Spot Assay System (V-PLEX Proinflammatory Panel 1 Human Kit, catalogue number K15049D, Meso Scale Diagnostics, Rockville, MA, USA) [66]. All reagents were equilibrated to room temperature prior to the assay. Calibration solutions were prepared through a series of four-fold dilution steps, and a zero calibrator was also established. CSF samples, plasma samples, and calibration solutions were pipetted onto MSD plates, which were subsequently incubated at room temperature with shaking for two hours. Following this initial incubation, plates were washed and a detection antibody solution was added. Plates were then incubated again at room temperature with shaking for another two hours. After the second incubation, plates were washed and read on the MSD instrument. Concentrations were determined based on calibration curves generated from the known calibrators.

### 4.4. CSF/Serum Albumin Quotient

Albumin in matched CSF and serum samples was measured on a Beckman Coulter (Brea, CA, USA) clinical chemistry analyser using a timed endpoint method assay which relies on the reaction of albumin with Bromocresol Purple to form a coloured complex; absorbance was measured at 600 nm. The CSF/serum albumin quotient (qAlb) was calculated as a proxy measure for BBB impairment [67,68].

### 4.5. Haptoglobin and Haemoglobin Analysis

Hp phenotype was determined by non-denaturing Western blot using 1:5000 polyclonal rabbit anti-Hp antibody (Sigma, Gillingham, Dorset, UK). CSF Hp and haemoglobin were assayed using an in-house UPLC technique combining size exclusion chromatography with eluate absorbance measurement at 415 nm [69]. CSF was run neat to quantify haemoglobin and then run again after the addition of a saturating quantity of haemoglobin to quantify Hp. To quantify haemoglobin, a standard curve (nine data points from 0 to 1 g/L) was prepared from lyophilized human haemoglobin (Sigma-Aldrich, Gillingham, Dorset, UK) reconstituted to 1 g/L in diluent (9 g/L NaCl, 10 mM EDTA). The concentration of the standard haemoglobin solution was verified independently by spectrophotometric quantification using HemocueTM (Hemocue, Ängelholm, Sweden). Serum Hp was assayed using immunoturbidimetry on a Beckman Coulter (Brea, CA, USA) clinical chemistry analyser. The CSF/serum Hp quotient (qHp) was calculated and used to determine the Hp index $\left(qHp÷qAlb\right)$.

### 4.6. Clinical Data and Outcome Measures

The following baseline variables were prospectively collected and available for analysis for all SAH participants: age, sex, race, premorbid history of hypertension, location of aneurysm, method of securing aneurysm, initial blood volume on the computed tomography scan, location of blood, and WFNS grade.

Whilst in hospital, patients were monitored clinically for delayed cerebral ischemia (DCI). Functional outcomes were assessed at days 90, and 180 with the CLCE-24 [30], GOSE [31], mRS [32], and SAHOT [33]. SAHOT raw scores were transformed to ordinal categories using the nomogram from the previously published Rasch analysis [33].

### 4.7. Statistical Methods

Descriptive statistics were used to summarise demographic and clinical characteristics and outcomes. Variables were tested for normality with the Shapiro–Wilks test. Means and confidence intervals are reported for parametric data. Medians and interquartile ranges (IQR) are reported for non-parametric data. The Kruskal–Wallis test and Dunn’s post hoc test were used to compare medians between independent groups. A comparison of proportions between groups was made with Fisher’s exact test. Linear regressions were used to determine the relationship between continuous variables. Principal component analysis was used to reduce the dimensionality of the CSF cytokines and examine the primary central inflammatory components. Moderation analysis was conducted to determine whether the relationship between serum Hp and cytokine levels varied as a function of Hp phenotype. Univariate logistic regression was used to examine the association between each outcome and Hp levels or type. mRS and SAHOT were analysed using ordinal logistic regression. *p* < 0.05 was deemed to be statistically significant. All analyses were performed in STATA/IC 16.1.

## 5. Conclusions

In summary, this study of the Hp response after SAH has three important and novel findings. First, SAH patients have a significantly lower Hp index compared to controls, suggesting that intrathecal Hp consumption occurs following SAH. Second, we have provided evidence that the Hp phenotype significantly affects intrathecal Hp turnover. Third, the significant association between CSF cytokine levels and serum Hp upregulation following SAH provides human evidence of a systemic acute phase response to a central inflammatory stimulus. These findings have important implications for our understanding of the pathophysiology of SAH and may inform the development of novel therapeutic interventions to improve outcomes for SAH patients, including intrathecal haptoglobin administration and therapies targeting EVs.

## Figures and Tables

**Figure 1 ijms-24-16922-f001:**
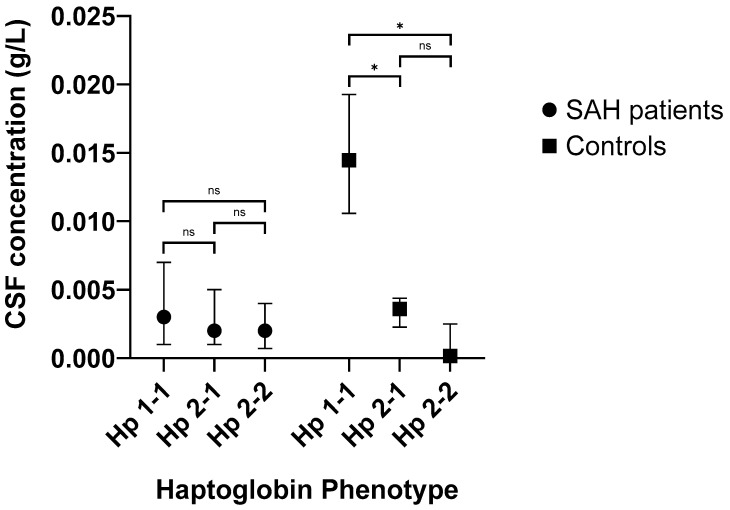
The levels of cerebrospinal fluid (CSF) haptoglobin (Hp) by Hp phenotype in subarachnoid haemorrhage (SAH) patients and controls. * *p* < 0.05; ns = not significant.

**Figure 2 ijms-24-16922-f002:**
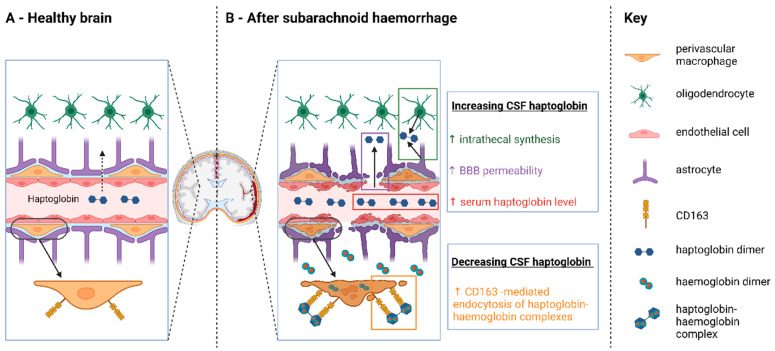
Factors influencing cerebrospinal fluid (CSF) haptoglobin concentration in the healthy state (**A**) and after SAH (**B**). CSF haptoglobin would increase as a result of higher serum haptoglobin and increased blood–brain barrier permeability, as well as upregulation of intrathecal haptoglobin synthesis. CSF haptoglobin would decrease as a result of intrathecal CD163-mediated uptake of haemoglobin–haptoglobin complexes after SAH. The intrathecal haptoglobin index corrects for serum haptoglobin levels and blood–brain barrier permeability, so that it becomes a measure of the balance between intrathecal synthesis and CD163-mediated consumption of haptoglobin. CSF = cerebrospinal fluid. BBB = blood–brain barrier.

**Figure 3 ijms-24-16922-f003:**
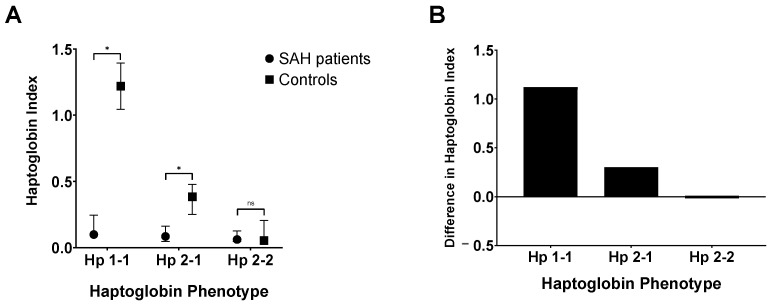
(**A**) Haptoglobin (Hp) index by Hp phenotype in subarachnoid haemorrhage (SAH) patients and controls. (**B**) Difference in Hp index between controls and patients (positive values represent a higher Hp index in controls). * *p* < 0.05; ns = not significant.

**Figure 4 ijms-24-16922-f004:**
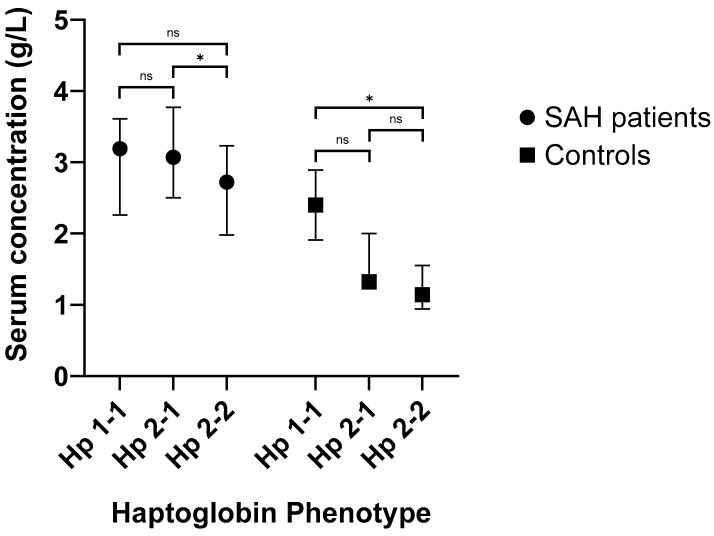
The levels of serum haptoglobin (Hp) by Hp phenotype in subarachnoid haemorrhage (SAH) patients and controls. * *p* < 0.05; ns = not significant.

**Figure 5 ijms-24-16922-f005:**
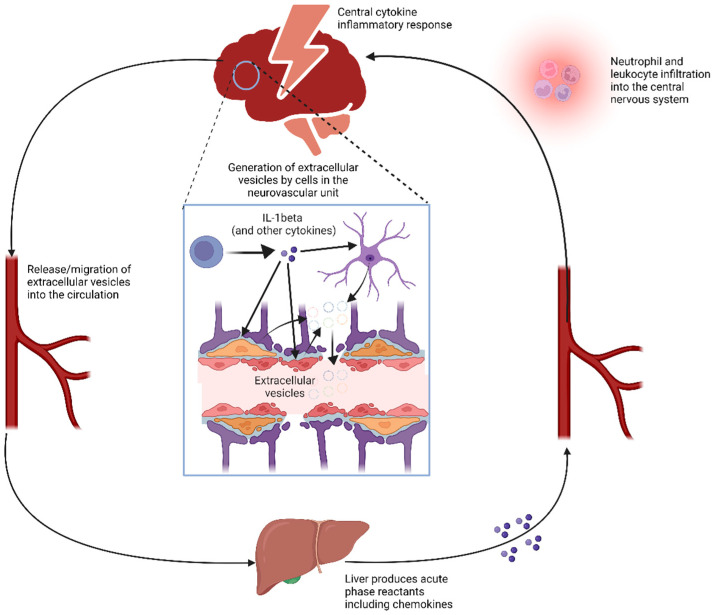
The liver acute phase response to central inflammation.

**Figure 6 ijms-24-16922-f006:**
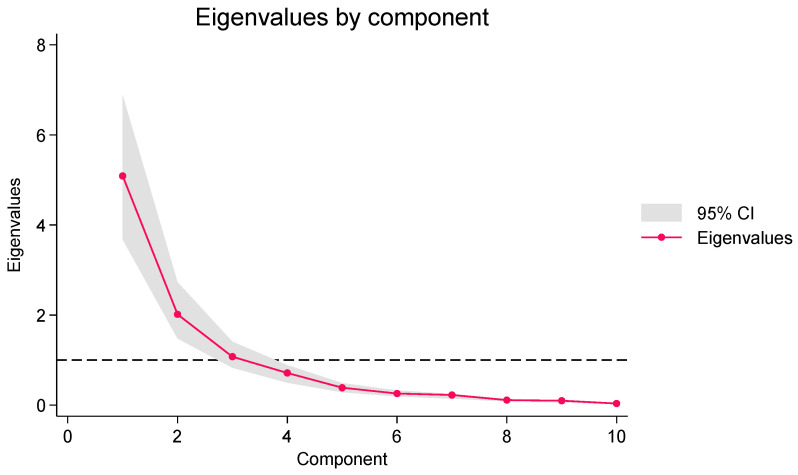
Scree plot of the principal components for cerebrospinal fluid cytokines. The horizontal dotted line represents an eigenvalue of 1.

**Figure 7 ijms-24-16922-f007:**
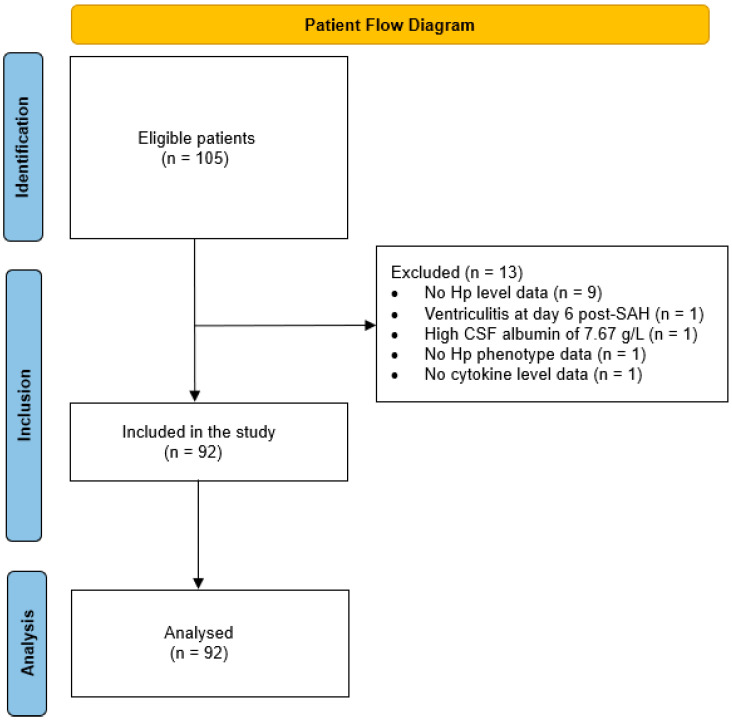
Patient flow diagram of participant inclusion and exclusion.

**Table 1 ijms-24-16922-t001:** Baseline characteristics by haptoglobin phenotype. ^a^ Fisher’s exact. ^b^ Kruskal–Wallis. World Federation of Neurosurgical Societies = WFNS; external ventricular drain = EVD.

		Hp 1-1 (n = 19)	Hp 2-1 (n = 46)	Hp 2-2 (n = 27)	*p*-Value
Age [median, IQR]		56 (48–62)	55 (49–59)	55 (51–68)	0.707 ^b^
Sex (n, %)	Female	14 (73.7)	33 (71.7)	22 (81.5)	0.663 ^a^
	Male	5 (26.3)	13 (28.3)	6 (18.5)	
Race (n, %)	White	19 (100.0)	46 (100.0)	24 (88.9)	0.082 ^a^
	Black	0 (0.0)	0 (0.0)	1 (3.7)	
	Asian	0 (0.0)	0 (0.0)	2 (7.4)	
Premorbid hypertension	Yes	6 (31.6)	11 (23.9)	9 (33.3)	0.592 ^a^
	No	13 (68.4)	35 (76.1)	18 (66.7)	
WFNS Grade (n, %)	I	8 (42.1)	20 (43.5)	13 (44.4)	0.985 ^a^
	II	3 (15.8)	9 (19.6)	4 (14.8)	
	III	2 (10.5)	2 (4.4)	3 (11.1)	
	IV	5 (26.3)	12 (26.1)	7 (25.9)	
	V	1 (5.3)	3 (6.5)	1 (3.7)	
Blood volume (n = 91) (cm^3^) [median, IQR]		11.8 (4.2–29.0)	21.2 (8.0–39.6)	24.3 (10.1–32.4)	0.367 ^b^
Intracerebral haemorrhage present (n, %)	Yes	1 (5.3)	6 (13.0)	7 (25.9)	0.179 ^a^
	No	17 (89.5)	40 (87.0)	20 (74.1)	
	Missing	1 (5.3)	0 (0.0)	0 (0.0)	NA
Intraventricular haemorrhage present (n, %)	Yes	14 (73.7)	32 (69.6)	23 (85.2)	0.340 ^a^
	No	4 (21.0)	14 (30.4)	4 (14.8)	
	Missing	1 (5.3)	0 (0.0)	0 (0.0)	NA
EVD inserted (n, %)	Yes	9 (47.4)	16 (34.8)	7 (25.9)	0.326 ^a^
	No	10 (52.6)	30 (65.2)	20 (74.1)	
Aneurysm location (n, %)	Anterior Cerebral	5 (26.3)	21 (45.7)	10 (37.0)	0.391 ^a^
	Internal Carotid	6 (31.6)	4 (8.7)	7 (25.9)	
	Middle Cerebral	5 (26.3)	14 (30.4)	6 (22.2)	
	Vertebrobasilar	3 (15.8)	6 (13.0)	3 (11.1)	
	Non-aneurysmal	0 (0.0)	1 (2.2)	1 (3.7)	
Securing of the aneurysm (n, %)	Clipping	4 (21.1)	11 (23.9)	6 (22.2)	1.000 ^a^
	Coiling	15 (78.9)	34 (73.9)	20 (74.1)	
	Not applicable	0 (0.0)	1 (2.2)	1 (3.7)	

**Table 2 ijms-24-16922-t002:** Association between serum haptoglobin levels and cerebrospinal fluid cytokines of subarachnoid haemorrhage patients. Moderation analysis is also presented to investigate whether the relationship between serum haptoglobin and cytokine levels varies as a function of haptoglobin phenotype (bolded for significant results).

Cytokines (pg/mL)	Variables	β (95% CI)	*p* Value
IFN-gamma	Serum Hp level	**11.591 (0.055–23.128)**	**0.049**
	Hp 2-1 (relative to Hp 1-1)	22.577 (−17.964–63.119)	0.271
	Hp 2-2 (relative to Hp 1-1)	**50.113 (7.011–93.215)**	**0.023**
	Interaction with Hp level and Hp 2-1	−9.729 (−22,420–2962)	0.113
	Interaction with Hp level and Hp 2-2	**−15.987 (−30.198–−1.776)**	**0.028**
IL-1beta	Serum Hp level	**1.413 (0.116–2.710)**	**0.033**
	Hp 2-1 (relative to Hp 1-1)	1.589 (−2.968–6.146)	0.489
	Hp 2-2 (relative to Hp 1-1)	**5.470 (0.625–10.315)**	**0.027**
	Interaction with Hp level and Hp 2-1	−0.598 (−2.025–0.828)	0.406
	Interaction with Hp level and Hp 2-2	**−1.725 (−3.322–−0.127)**	**0.035**
IL-10	Serum Hp level	1.271 (−3.917–6.458)	0.627
	Hp 2-1 (relative to Hp 1-1)	2.120 (−16.110–20.350)	0.817
	Hp 2-2 (relative to Hp 1-1)	18.501 (−0.880–37.883)	0.061
	Interaction with Hp level and Hp 2-1	−0.723 (−6.430–4.984)	0.801
	Interaction with Hp level and Hp 2-2	−5.227 (−11.617–1.164)	0.107
IL-12p70	Serum Hp level	**9.796 (2.787–16.806)**	**0.007**
	Hp 2-1 (relative to Hp 1-1)	17.235 (−7.397–41.867)	0.167
	Hp 2-2 (relative to Hp 1-1)	**28.436 (2.248–54.624)**	**0.034**
	Interaction with Hp level and Hp 2-1	−7.208 (−14.918–0.503)	0.066
	Interaction with Hp level and Hp 2-2	**−9.518 (−18.152–−0.884)**	**0.031**
IL-13	Serum Hp level	9.833 (−2.544–22.210)	0.118
	Hp 2-1 (relative to Hp 1-1)	−5.668 (−49.164–37.829)	0.796
	Hp 2-2 (relative to Hp 1-1)	**49.495 (3.252–95.739)**	**0.036**
	Interaction with Hp level and Hp 2-1	−0.481 (−14.097–13.135)	0.944
	Interaction with Hp level and Hp 2-2	**−18.042 (−33.289–−2.795)**	**0.021**
IL-2	Serum Hp level	**1.214 (0.064–2.365)**	**0.039**
	Hp 2-1 (relative to Hp 1-1)	1.584 (−2.460–5.627)	0.438
	Hp 2-2 (relative to Hp 1-1)	**5.498 (1.199–9.797)**	**0.013**
	Interaction with Hp level and Hp 2-1	−0.654 (−1.919–0.612)	0.307
	Interaction with Hp level and Hp 2-2	**−1.800 (−3.217–−0.382)**	**0.014**
IL-4	Serum Hp level	**8.570 (2.683–14.458)**	**0.005**
	Hp 2-1 (relative to Hp 1-1)	14.054 (−6.634–34.743)	0.180
	Hp 2-2 (relative to Hp 1-1)	**26.247 (4.252–48.242)**	**0.020**
	Interaction with Hp level and Hp 2-1	**−6.243 (−12.719–0.234)**	**0.059**
	Interaction with Hp level and Hp 2-2	**−8.675 (−15.927–−1.423)**	**0.020**
IL-6	Serum Hp level	**3057.935 (938.419–5177.451)**	**0.005**
	Hp 2-1 (relative to Hp 1-1)	5570.533 (−1877.866–13,018.93)	0.140
	Hp 2-2 (relative to Hp 1-1)	**9021.475 (1102.616–16,940.33)**	**0.026**
	Interaction with Hp level and Hp 2-1	**−2391.482 (−4723.056–−59.908)**	**0.045**
	Interaction with Hp level and Hp 2-2	**−2988.535 (−559−9.454–−377.617)**	**0.025**
IL-8	Serum Hp level	408.761 (−567.051–1384.573)	0.407
	Hp 2-1 (relative to Hp 1-1)	230.141 (−3199.058–3659.339)	0.894
	Hp 2-2 (relative to Hp 1-1)	2266.3 (−1379.495–5912.095)	0.219
	Interaction with Hp level and Hp 2-1	84.080 (−989.363–1157.523)	0.876
	Interaction with Hp level and Hp 2-2	−491.748 (−1693.799–710.303)	0.418
TNF-alpha	Serum Hp level	2.023 (−0.899–4.945)	0.172
	Hp 2-1 (relative to Hp 1-1)	1.579 (−8.690–11.848)	0.760
	Hp 2-2 (relative to Hp 1-1)	**14.599 (3.681–25.517)**	**0.009**
	Interaction with Hp level and Hp 2-1	−0.507 (−3.722–2.707)	0.754
	Interaction with Hp level and Hp 2-2	**−4.730 (−8.305–−1.155)**	**0.011**
Principal Component 1	Serum Hp level	**1.850 (0.580–3.120)**	**0.005**
	Hp 2-1 (relative to Hp 1-1)	2.460 (−2.002–6.921)	0.276
	Hp 2-2 (relative to Hp 1-1)	**7.520 (2.776–12.264)**	**0.002**
	Interaction with Hp level and Hp 2-1	−1.041 (−2.438–0.355)	0.142
	Interaction with Hp level and Hp 2-2	**−2.428 (−3.992–−0.864)**	**0.003**
Principal Component 2	Serum Hp level	−0.482 (−1.339–0.375)	0.266
	Hp 2-1 (relative to Hp 1-1)	−1.546 (−4.557–1.464)	0.309
	Hp 2-2 (relative to Hp 1-1)	−0.011 (−3.212–3.190)	0.995
	Interaction with Hp level and Hp 2-1	0.702 (−0.240–1.645)	0.142
	Interaction with Hp level and Hp 2-2	0.070 (−0.986–1.125)	0.896
Principal Component 3	Serum Hp level	−0.078 (−0.724–0.569)	0.812
	Hp 2-1 (relative to Hp 1-1)	0.028 (−2.244–2.300)	0.980
	Hp 2-2 (relative to Hp 1-1)	0.847 (−1.569–3.263)	0.487
	Interaction with Hp level and Hp 2-1	0.040 (−0.672–0.751)	0.912
	Interaction with Hp level and Hp 2-2	−0.170 (−0.967–0.626)	0.671

**Table 3 ijms-24-16922-t003:** Heat map of the eigenvectors that form each component of the cerebrospinal fluid cytokine principal components. Green represents an eigenvector of magnitude ≥ 0.3; yellow represents an eigenvector of magnitude ≥ 0.1 and <0.3; red represents an eigenvector of magnitude < 0.1.

Variable	Component 1	Component 2	Component 3	Unexplained
CSF IFN-gamma	0.2979	−0.1914	−0.1327	0.4555
CSF IL-1beta	0.3856	0.1303	0.3081	0.1071
CSF IL-10	0.1334	0.0327	0.8927	0.05044
CSF IL-12p70	0.3444	−0.3718	−0.064	0.1134
CSF IL-13	0.3426	0.2788	−0.2138	0.1967
CSF IL-2	0.3455	0.1562	−0.1825	0.3075
CSF IL-4	0.3479	−0.4017	−0.0369	0.05734
CSF IL-6	0.331	−0.3883	0.0043	0.1386
CSF IL-8	0.2652	0.4453	−0.0242	0.2419
CSF TNF-alpha	0.2975	0.441	−0.0744	0.1519

## Data Availability

The data presented in this study are available on request subject to ethical, funder and institutional approvals.

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
