# Peer review of "The Haptoglobin Response after Aneurysmal Subarachnoid Haemorrhage"

_ijms, 2023, doi:10.3390/ijms242316922_

Round 1
Reviewer 1 Report
Comments and Suggestions for Authors Bandyopadhyay et al are presenting a very interesting study aiming to evaluate the haptoglobin response after aneurysmal subarachnoid haemorrhage. The manuscript is well written and easy to follow. The scientific interest is unquestionable. I do have some minor points to improve the paper : 1. Define all acronyms at their first presentation: "especially in HP1 allele carriers" "haptoglobin (qHp) - line 13 not 18 2. How were the controls selected and included 3. Replace STROBE Flow Diagram by Patient Flow diagram , otherwise the authors should be more rigorous in following the STROBE recommendations for reporting observational studiesAuthor Response
Reviewer 1
Bandyopadhyay et al are presenting a very interesting study aiming to evaluate the haptoglobin response after aneurysmal subarachnoid haemorrhage. The manuscript is well written and easy to follow. The scientific interest is unquestionable.
Response: Thank you for the kind words about the value of this study.
I do have some minor points to improve the paper :
- Define all acronyms at their first presentation: "especially in HP1 allele carriers" "haptoglobin (qHp) - line 13 not 18
Response: We have now ensured that the abstract uses the unabbreviated form of “haptoglobin”, and that the main text continues to use the abbreviated form of haptoglobin (Hp) only after the first use of the word and explaining the abbreviation. We have amended the line with qHp on page 2 to explain that the acronym relates to the ratio of CSF/serum haptoglobin.
- How were the controls selected and included
Response: We have added details about where the controls were selected from, in the Methods section on page 4, and why they were selected.
- Replace STROBE Flow Diagram by Patient Flow diagram, otherwise the authors should be more rigorous in following the STROBE recommendations for reporting observational studies
Response: Thank you for this pertinent observation. This is not an observational cohort study, but a secondary use of samples and data collected during a randomised controlled trial. We have better clarified this in the Methods section on page 4, and also relabelled the Flow Diagram as suggested.
Reviewer 2 Report
Comments and Suggestions for Authors
The present study about the haptoglobin response after aneurysmal SAH is quite original, on the other hand few problems are encountered as follow:
1) It is not at all clear the study is about what:1) to investigate the haptoglobin response or 2) to investigate the safety and efficacy of SFX-01? (please explain, add details)
2) It is very hard to understand why the patients and methods section is placed after the discussion section and not before the results (please explain, add details and if possible amend).
3) I wonder how potential readers could understand the study setting and development without reporting the study protocol itself? (Please add, explain)
4) To evaluate the scientific value of a study and especially of a randomized double blind one it is crucial to state the primary and secondary (if any) study endpoints that unfortunately are missing, (Please add)
5) The discussion and conclusion section are actually missing the therapeutic perspective (please add few lines, develop).
6) Pertinent and relevant literature is missing as follow: Messina R, de Gennaro L, De Robertis M, Pop R, Chibbaro S, Severac F, Blagia M, Balducci MT, Bozzi MT, Signorelli F. Cerebrospinal Fluid Lactate and Glucose Levels as Predictors of Symptomatic Delayed Cerebral Ischemia in Patients with Aneurysmal Subarachnoid Hemorrhage. World Neurosurg. 2023 Feb;170:e596-e602. doi: 10.1016/j.wneu.2022.11.068. Epub 2022 Nov 18. PMID: 36403937.
Finally I do believe that the study being quite original needs to be globally reorganized.
Author Response
Reviewer 2
The present study about the haptoglobin response after aneurysmal SAH is quite original,
Response: Thank you for the kind words about the value of this study.
on the other hand few problems are encountered as follow:
1) It is not at all clear the study is about what:1) to investigate the haptoglobin response or 2) to investigate the safety and efficacy of SFX-01? (please explain, add details)
Response: We have revised the manuscript on page 3 and page 4 to clarify that the objective of this study was not to study the safety and efficacy of SFX-01, but to use the samples and data collected in this randomised controlled trial to investigate the haptoglobin response after subarachnoid haemorrhage.
2) It is very hard to understand why the patients and methods section is placed after the discussion section and not before the results (please explain, add details and if possible amend).
Response: We completely agree with you and believe the Methods section should be before the Results section, but we were asked to revise the submission by the editorial team and place the Methods section after the Results. We have therefore resubmitted our original manuscript file with the Methods section preceding the Results, but it is possible that editorial policy does not allow this.
3) I wonder how potential readers could understand the study setting and development without reporting the study protocol itself? (Please add, explain)
Response: Thank you for this comment. The protocol for the original trial can be found in reference 28.
4) To evaluate the scientific value of a study and especially of a randomized double blind one it is crucial to state the primary and secondary (if any) study endpoints that unfortunately are missing, (Please add)
Response: We have added the primary and secondary endpoints towards the end of the Introduction on page 3 and page 4. As stated above, we have clarified on page 4 that this study used samples and data collected in a randomised controlled trial to investigate the haptoglobin response after subarachnoid haemorrhage.
5) The discussion and conclusion section are actually missing the therapeutic perspective (please add few lines, develop).
Response: We have added more sentences on page 17 and page 18 to relate the study to the therapeutic perspective in the Discussion and Conclusion.
6) Pertinent and relevant literature is missing as follow: Messina R, de Gennaro L, De Robertis M, Pop R, Chibbaro S, Severac F, Blagia M, Balducci MT, Bozzi MT, Signorelli F. Cerebrospinal Fluid Lactate and Glucose Levels as Predictors of Symptomatic Delayed Cerebral Ischemia in Patients with Aneurysmal Subarachnoid Hemorrhage. World Neurosurg. 2023 Feb;170:e596-e602. doi: 10.1016/j.wneu.2022.11.068. Epub 2022 Nov 18. PMID: 36403937.
Response: We have added the topic of this paper to the Discussion on page 18, and referenced it as requested.
Finally I do believe that the study being quite original needs to be globally reorganized.
Response: Thank you for your suggestions. We hope that by making the above changes, we have improved the organization of the paper.
Round 2
Reviewer 2 Report
Comments and Suggestions for Authors
The authors answered satisfactorily to reviewers comments, raised criticisms and suggestions. The manuscript has been globally improved and it now could be considered for publication.